# Using Behavior Integration to Identify Barriers and Motivators for COVID-19 Vaccination and Build a Vaccine Demand and Confidence Strategy in Southeastern Europe

**DOI:** 10.3390/vaccines12101131

**Published:** 2024-10-02

**Authors:** Stefan Mandić-Rajčević, Smiljana Cvjetković, Lisa Oot, Dalibor Tasevski, Ankita Meghani, Hannah Wallace, Tatiana Cotelnic, Dragoslav Popović, Elan Ebeling, Tonja Cullen Balogun, Lynne Cogswell

**Affiliations:** 1JSI, Arlington, VA 22202, USA; smiljana.cvjetkovic@med.bg.ac.rs (S.C.); lisa_oot@jsi.com (L.O.); drdako2000@gmail.com (D.T.); hannah_wallace@jsi.com (H.W.); tonja_cullen-balogun@jsi.com (T.C.B.); lynne_cogswell@jsi.com (L.C.); 2Institute of Social Medicine and School of Public Health and Healthcare Management, Faculty of Medicine, University of Belgrade, 11000 Belgrade, Serbia; 3Department of Humanities, Faculty of Medicine, University of Belgrade, 11000 Belgrade, Serbia; 4PATH, Seattle, WA 98121, USA; ameghani@path.org (A.M.); elan_ebeling@jsi.com (E.E.); 5Center for Health Policies and Studies, MD-2012 Chisinau, Moldova; tatiana.cotelnic@pas.md; 6Serbian Public Health Association, 11000 Belgrade, Serbia; popovicteam@gmail.com

**Keywords:** Behavior Integration, COVID-19 vaccines, vaccine hesitancy, health behavior, community engagement, health promotion, Southeastern Europe

## Abstract

**Introduction.** The COVID-19 pandemic has significantly impacted global health, with Eastern Europe experiencing notable excess morbidity and mortality and vaccine hesitancy. This study utilized the Behavior Integration (BI) framework to identify barriers and motivators for COVID-19 vaccination and develop strategies to increase vaccine demand and confidence in Moldova, North Macedonia, and Serbia. **Methods.** A mixed-methods approach was employed, including qualitative interviews and quantitative surveys. The BI framework was used to integrate human behaviors with technical and operational considerations throughout the project. **Results.** A total of 2742 online surveys were collected in Moldova and Serbia, revealing significant barriers such as vaccine safety concerns (OR = 1.839, 95% CI: 1.328–2.547 in urban Moldova; OR = 1.990, 95% CI: 1.351–2.931 in urban Serbia), logistical challenges, and lack of institutional trust. Conversely, motivators included personal health concerns, recommendations from health care providers, and the desire to travel. The proposed social and behavior change strategy included a continuing medical education course that trained 2403 medical providers, with post-test results showing a 99% improvement in knowledge and confidence in applying the information, and collective engagement workshops for 3260 chronic disease patients and 842 pregnant women, of which 7% were vaccinated against COVID-19. **Conclusions.** The BI approach effectively identified and addressed vaccination barriers and motivators, leading to tailored strategies that increased vaccine uptake. Continuous stakeholder engagement, adaptive learning processes, and local organizations are crucial for refining program implementation, ensuring sustainability, and promoting public health.

## 1. Introduction

As of May 2024, there had been more than 775 million confirmed cases of COVID-19 in the world, and more than 7 million people had lost their lives due to this disease [1]. In Europe, there were about 251 million confirmed cases and 2.08 million confirmed deaths from COVID-19 [2,3]. In Europe, the pandemic resulted in substantial economic costs, hindering delivery of health care services and leading to changes in the epidemiology and treatment of many acute conditions, which exacerbated health disparities [4,5]. Along with economic issues, the pandemic has also posed ethical challenges, where one of the key questions is how to achieve the common good. The principle of the common good implies that what is good for society is also good for the individual. Furthermore, from the global health perspective that means that patients cannot be perceived as isolated human beings but persons strongly bound with their communities, suggesting that both patients and the community need to be taken into consideration [6].

As of March 2022, estimates for COVID-19 vaccination coverage in Europe ranged from 65% to 68%, but coverage varied greatly, with Southeastern European and Balkan countries lagging behind the European Union [7,8]. At the same time, just two months before the start of the MOMENTUM project, WHO estimated only 26.9% of Moldova’s population had received at least one dose. In North Macedonia, only 40.8% of the target population (health care workers, people in long-term care facilities, older adults, and people with underlying health conditions) had received a full course of COVID-19 vaccination (defined as two doses). In Serbia, COVID-19 vaccines first became available in December 2020, and the country prioritized the aforementioned populations. These efforts resulted in the vaccination of 69.4% of the country’s health workers and 48.8% of the population receiving at least one dose by March 2022. The United States Agency for International Development (USAID)-funded MOMENTUM Routine Immunization Transformation and Equity (the project) began work in the Southeastern European countries of Moldova, North Macedonia, and Serbia in May 2022.

During the COVID-19 pandemic, Eastern Europe experienced a higher prevalence of vaccine skepticism than other parts of Europe due in large part to reports of distrust in national governments and medical professionals as sources of relevant medical information [9]. The main determinants of COVID-19 vaccine hesitancy in Eastern European countries include factors such as public confidence in vaccine safety and efficacy, vaccine literacy, and trust in the government and medical system [8]. Factors that have hindered vaccine acceptance in the Southeastern European countries of Moldova, North Macedonia, and Serbia include societal distrust in health and political authorities, perceptions of profit motives behind vaccine development, and low institutional trust [10,11,12]. The low vaccination rates may also be attributed to the failure of institutions to communicate the benefits of vaccination, particularly to those with comorbidities and who fear side effects [8]. In North Macedonia, there is a lower level of societal trust, with factors such as younger age and lower health literacy predicting this [12]. Serbians are less confident that political authorities recommend vaccines with the best intentions than people elsewhere in the region [10,11].

Globally, various models have been used to study vaccine behavior, hesitance, and acceptance. The Health Belief Model, one of the most commonly used in health and illness behavior studies, comprises three primary constructs: perceived benefits, perceived barriers, and cues to action [13,14]. The 5C Model of vaccine hesitancy evaluates individual determinants (confidence, complacency, convenience, risk calculation, and collective responsibility) to understand causes of vaccine hesitancy or acceptance [15]. The Theory of Planned Behavior has been used to predict the general population’s intention to receive the COVID-19 vaccine [16]. Contrary to behavior models, behavior *change* models focus on guiding individuals through stages of behavior change and intervention strategies to facilitate behavior change. These models, including Behavior Integration (BI), incorporate elements of behavior models but are more focused on practical strategies for modifying behavior, including stages of readiness, barriers, and motivators to adopting new behaviors.

Despite the growing interest in health behavior models and behavior change theories, there is a significant lack of implementation research that follows the entire process of identifying behaviors, their drivers, barriers, and motivators, implementing strategies, and measuring their effect. Although behavior change theories can inform implementation strategies, they are often inadequately applied and documented in practice, resulting in suboptimal outcomes [17]. Evaluation of interventions aimed at changing health professionals’ behavior usually comes from retrospective studies that are unable to capture the complexities and context of implementation [18]. Finally, there is an important gap in understanding how to effectively implement interventions targeting multiple behaviors. This gap is particularly evident in the absence of qualitative research and adaptive learning processes, which are essential for refining strategies based on real-time insights and context-specific factors [19]. Without focused implementation research, including qualitative studies and adaptive learning, the potential for successful health behavior change remains significantly underutilized.

This paper aims to achieve the following: (1) present and apply BI to identify COVID-19 vaccination barriers and motivators; (2) detail how BI was used to develop a program strategy to increase vaccine demand and confidence; and (3) present the results and learning of this strategy on COVID-19 vaccine uptake in Moldova, North Macedonia, and Serbia.

## 2. Materials and Methods

### 2.1. Setting

USAID’s Europe and Eurasia Regional Bureau asked the project to accelerate widespread and equitable access to and delivery of safe and effective COVID-19 vaccinations by increasing demand for and uptake of COVID-19 vaccination among priority populations in Moldova, North Macedonia, and Serbia. The project’s scope was the following: (1) increase demand for COVID-19 vaccination; (2) dispel myths and mis/disinformation; (3) implement behavior change interventions; and (4) share lessons and promising practices within and between Moldova, North Macedonia, and Serbia and across the region by using BI to develop tailored health professional capacity building and collective engagement (CE) programming in each country (although not part of project implementation, stakeholders from Bosnia and Herzegovina were included in the project’s regional learning exchanges).

### 2.2. Behavior Integration (BI)

BI is a behavior change framework that ensures human behaviors are integrated alongside technical and operational considerations throughout every phase of program development. The BI approach uses local evidence (data) to design strategies and allows for iteration and adaptation as part of the process. Furthermore, BI emphasizes behavior-led over intervention-driven strategies and complements traditional social and behavior change methodologies such as human-centered design and systems strengthening. The BI process includes the steps shown in Figure 1. ThinkBIG is a set of procedures and tools for applying BI to increase the uptake and practice of priority behaviors (Box 1).

Box 1Overview of the Behavioral Integration Approach.Behavior Integration (https://www.jsi.com/behavior-integration/, accessed 1 October 2024) defines outcomes as specific behaviors required to achieve a development goal [20]. It ensures that behavior is considered along with technical and operational issues in every program element and phase. The Manoff Group, acquired by JSI in 2022, co-created Behavior Integration Guidance—ThinkBIG (https://thinkbigonline.org/tools, accessed 1 October 2024) under the USAID-funded ACCELERATE Project (2015–2020) as a resource for applying BI to USAID-funded projects and processes and increasing the uptake and practice of priority behaviors to reduce maternal and child mortality.

Identifying a program goal, analyzing causes preventing the achievement of the goal, identifying and prioritizing behaviors, and mapping pathways to change are informed by a formative assessment (see Section 2.3). Behavior profiles are developed by identifying steps needed to practice the priority behaviors; defining factors that prevent or support the practice of the behavior; and identifying supporting actors and developing strategies (Figure 2). When mapping pathways to change, the first step is to analyze factors, both barriers and motivators, divided into structural, social, and internal to the specific behavior (Appendix A). The second step connects each barrier and motivator with a specific supporting actor and his/her action, which could either reduce/remove the barrier or increase/strengthen/create a motivator. Each actor creates a network of support that is essential for integrating and sustaining behavior change initiatives (Appendix A).

### 2.3. Analyzing Root Causes through Formative Assessment

Between September and December 2022, the project conducted a mixed-methods formative assessment to identify barriers of people in Moldova, North Macedonia, and Serbia to getting vaccinated and what motivators to do so.

#### 2.3.1. Qualitative Data Collection

Qualitative data were collected through 21 in-depth interviews in Moldova, North Macedonia, and Serbia. Purposeful sampling was applied, and stakeholders were selected based on their professional experience and particular knowledge, which allowed a deeper insight into all the aspects relevant to this study. The interviewees represented the following: (1) national and regional institutes of public health; (2) ministries of health; (3) professional health organizations such as the Chamber of Medical Doctors and Chamber of Nurses; (4) professional media organizations; and (5) scientists affiliated with universities and nongovernmental health organizations (Appendix A). The interviews aimed to achieve the following: (1) understand specific barriers and motivators to COVID-19 vaccinations and dispel misconceptions and myths; (2) identify vaccine-hesitant priority groups; (3) describe the successes and challenges of each country’s strategies to enhance COVID-19 vaccination demand and uptake; (4) inform the design of social behavior change approaches; and (5) promote learning exchanges to overcome vaccination barriers among key priority groups.

The project audio recorded and wrote memos after each interview and analyzed these data thematically according to the Behavioral and Social Drivers framework. This framework outlines the factors that affect the overall uptake of the recommended vaccination: (1) thinking and feeling; (2) social processes; (3) motivation; and (4) practical issues [21].

#### 2.3.2. Quantitative Data Collection

In Moldova and Serbia, the project adapted questions from the validated Behavioral and Social Drivers for Vaccination toolkit to develop an online survey tool administered through the crowdsourcing platform PREMISE (https://premise.com/, accessed on 1 October 2024). The tool was piloted in both countries with 100 participants in each country. In the pilot, unvaccinated individuals (50) and health workers (25) were the main targets of the data collection. After revising the tool based on feedback from the pilot, the project collected data from 2742 participants living in Serbia and Moldova (note: at the time of data collection, PREMISE was not operational in North Macedonia, so data could not be collected there). This platform recruited participants via word of mouth and digital advertisements managed by Google directed at people in Moldova and Serbia, covering diverse socioeconomic characteristics and targeting the collection of information about potential key populations such as older people, those with chronic diseases, pregnant women, and health care workers. PREMISE invites people to join the platform and subsequently offers various surveys through an app. Upon completing surveys, participants receive an incentive, the specifics of which are proprietary. The questionnaire used for quantitative data collection was based on the Behavioral and Social Drivers tool [21] and ThinkBIG list of factors (Appendix A). The full questionnaire is available in Appendix A.

#### 2.3.3. Ethics Approval

This study was conducted in accordance with the Declaration of Helsinki and approved by the institutional review boards of JSI (approval 22-56 PJ1 AM1); Ministry of Health of Moldova; National Committee for Ethical Expertise of Clinical Trial (approval 1376 from 28 September 2022); Faculty of Medicine of the University of Skopje (approval 03-3933/1 from 19 September 2022); and the Faculty of Medicine, University of Belgrade (approval 1322/VII-25 from June 2022).

### 2.4. Development and Validation of Behavioral Profiles and Program Strategies

The project used the results of the formative assessment to create behavior profiles (pathways to change) and proposed strategies for validation by stakeholders from Moldova, North Macedonia, and Serbia.

#### 2.4.1. Validation of the Vaccine Demand and Confidence Strategy

The project used a participatory process to validate the formative assessment findings and the content outlined in each behavioral profile and to help prioritize and agree on final strategies for implementation. The process involved engaging public health officials, community leaders, and representatives from professional organizations in structured workshops and discussions. During these sessions, the project presented key research findings with the behavior profiles and pathways to change, leading to the development of behavior-led strategies. Stakeholders provided critical insights and identified gaps and areas for improvement. This process facilitated the refinement of the strategies by ensuring they were contextually appropriate and aligned with local needs and priorities. The participatory approach enhanced the research credibility and fostered a sense of ownership and commitment among participants, increasing the likelihood of successful implementation.

#### 2.4.2. Final Design

Through the BI process, the project prioritized and designed a comprehensive social behavior change strategy that focused on key demand generation strategies, with an overall focus on vaccination as an essential part of a healthy lifestyle. Formative assessment data indicated that health care providers needed more information and skills to communicate with patients, particularly higher-risk target populations, about COVID-19 vaccines. The project designed a continuing medical education (CME) course for providers who worked with pregnant women and older people living with chronic disease. The project worked with local health system partners to develop the course, which included a pre-test knowledge-based quiz; sessions on basic vaccinology, components of a healthy lifestyle including vaccination, and high-quality patient care; a post-test quiz; and an evaluation form. The course was accredited in Serbia and North Macedonia but not in Moldova, where it is used to build health care provider capacity.

In addition to building health care provider capacity to promote COVID-19 vaccination, the project simultaneously sought to increase demand and uptake among priority populations through workshops. In each country, the project worked with local community-based organizations using collective engagement (CE), a participatory learning approach that facilitates multiple interactions and mutual exchange of ideas and knowledge over a fixed period with multiple points of contact. During the workshops, attendees learned about healthy lifestyle behaviors and the importance of vaccination, including COVID-19, through handouts, stickers, puzzles, role plays, and case studies. At the end of the workshops, participants committed to taking a step toward COVID-19 vaccination, such as talking about it with a partner, making an appointment with their doctors to discuss vaccination, scheduling an appointment, and getting vaccinated. Following each workshop, participants received text messages and digital stickers reinforcing the key messages about healthy lifestyle behaviors.

Identified priority behaviors to reduce excess morbidity and mortality due to COVID-19 in Moldova, North Macedonia, and Serbia and links to full behavior profiles are available in Appendix A. The comprehensive strategy determined by behavioral profiles is provided in Appendix A. It includes strategies which are wider than the scope of work of the project which was implemented.

### 2.5. Program Implementation

Across all three countries, the project implemented the CME course and CE activities with in-country partners, prioritizing the roll out of CME trainings before CE workshops. This progression increased provider capacity to support community members who attended the workshops. In Moldova, the National Agency for Public Health facilitated the CME course and worked with institutes of public health to invite health professionals to the trainings and recruit community members for the workshops. In North Macedonia, the community-based organization Emancipation, Solidarity and Equality of Women coordinated project activities and worked with the national immunization coordinator and members of the University of Skopje’s Medical Faculty to roll out the course. In Serbia, public health professionals of the Network of Public Health Institutes and the Medical and Pharmaceutical Faculties of Serbia participated in a training-of-trainers workshop. The collaboration also included the National Alliance for Local Economic Development and Institute of Public Health of Serbia “Dr. Milan Jovanović Batut” to roll out the CME trainings across the project districts. CE activities were further tested, adapted, coordinated, and implemented by facilitators from the community-based organization Association Rainbow. The CME and CE activities were implemented in Moldova from December 2023 to March 2024; North Macedonia from August 2023 to March 2024; and in Serbia from August 2023 to April 2024.

### 2.6. Program Monitoring, Evaluation, and Learning

Throughout program implementation, the project monitored the data across the three countries. It tracked the number of health care providers trained in the CME course and implemented pre- and post-tests to assess their COVID-19 vaccination knowledge. The post-test included an evaluation section that asked participants to indicate their confidence in applying knowledge gained from the course; if they learned new information; and for general input on the course quality.

The project conducted follow-up calls three weeks after the CE workshops to determine participants’ vaccination status. Several weeks after these calls, the team purposely selected two CE participants—one vaccinated and one unvaccinated—from each country to participate in an interview to further assess enablers of and barriers to vaccination (Appendix A). Additionally, every three weeks after workshops were implemented in each country until the project ended, the team assessed the following: (1) trust in information from health professionals and (2) perception of COVID-19 as a health concern as motivators of COVID-19-vaccine seeking behaviors.

The data collection forms were developed and stored in a web-based monitoring and evaluation system called LogAlto, which tabulated and exported frequencies to Excel as needed. In addition to quantitative data, the project sought qualitative feedback from CME trainers and CE facilitators during monthly pause-and-reflect meetings and quarterly learning exchanges. This allowed the project to continuously learn and iterate on the program strategy and design, aligned with principles of adaptive learning (https://usaidmomentum.org/resource/adaptive-learning-guide/, accessed 1 October 2024).

More broadly, the project aimed to deepen in-country partners’ understanding of adaptive learning and encouraged them to incorporate rapid feedback cycles into program implementation. These cycles involved brief interviews with CME trainers, CE facilitators, and participants. Participants were asked about which aspects of the trainings and workshops were most and least useful and to recommend ways to improve them. Facilitators were asked about which aspects of the event went well, challenges, and what they would do differently next time. The feedback was captured in a spreadsheet-based note-taking tool and synthesized (Appendix A). The project conducted fortnightly data review meetings with partner nongovernmental organizations to review the quantitative data (including performance monitoring plan indicators) and qualitative insights from rapid feedback. These meetings helped the project identify what went well and challenges and solutions.

## 3. Results

### 3.1. Formative Assessment Results

Data from the formative assessment informed the development of behavioral profiles and the eventual creation of behavior-led strategies. Main themes and subthemes emerging from the qualitative data collection were divided into barriers and motivators, as well as key actors/influencers (Appendix A). Key barriers included concerns about vaccine safety and efficacy, particularly among vulnerable groups like pregnant women, individuals with chronic diseases, and those planning to have children. Misconceptions about the rapid vaccine development process and recurring COVID-19 infections post-vaccination further fueled hesitancy. Younger individuals’ emphasis on personal freedom and autonomy, along with complacency due to the perceived low threat of COVID-19, also blocked vaccination efforts. Deeply ingrained misnomers about vaccination, such as fears that vaccination can cause infertility, and lack of trust in government and media, perceived as politically driven, compounded these issues, as did social factors, including widespread misinformation and insufficient public counter-messaging. Despite these barriers, several motivators were identified. Personal health concerns and the fear of contracting COVID-19 drove health-conscious individuals—especially those with chronic conditions like diabetes, who were often influenced by their health care providers’ recommendations—to get vaccinated. Doctors had a crucial role, with their recommendations significantly shaping decisions, although peer influence and anti-vaccine activists complicated this dynamic. The mass media’s focus on negative narratives and the failure of professional medical organizations to lead public communication inadvertently supported vaccine skepticism.

Quantitative data collection results are presented in detail in Appendix A. In Moldova, the study population consisted of 1412 participants (with valid surveys), with a slight majority being female (54.0%) compared to males (43.1%). The age distribution revealed that nearly half (47.2%) were between 18 and 25 years, while 23.5% were aged 26–35 years. Most participants resided in city centers or metropolitan areas (46.0%), with a significant portion from rural areas (34.6%). The education levels were diverse, with 39.4% holding secondary education and 30.0% possessing a university or college degree. Employment status indicated a large proportion of students (37.0%), followed by full-time employed individuals (24.6%). The predominant ethnic group was Moldovan (73.8%), with smaller representations of Russian (12.3%), Romani (4.0%), and Ukrainian (5.0%) minorities.

In Serbia, there were 1112 participants (with valid surveys), with females comprising the majority (60.2%) and males representing 38.8%. The largest age group was also 18–25 years (33.5%), followed by participants aged 26–35 years (26.0%). Nearly half (45.7%) resided in city centers or metropolitan areas, while 37.5% were from sub-urban or peri-urban areas. The majority had completed secondary education (61.8%), with 23.6% holding university or college degrees. Employment data showed 47.6% were full-time employed, and 22.1% were students. The vast majority of participants identified as Serbian (93.2%), with smaller ethnic groups such as Romani, Russian, and Bulgarian collectively making up less than 5%.

Because barriers and motivators of vaccination behavior often differ significantly between rural and urban populations, they were analyzed separately for Moldova and Serbia. Among barriers, safety concerns were significant in urban Moldova and urban and rural Serbia. Difficulty obtaining the preferred vaccine was not a significant barrier anywhere. Logistical challenges such as making appointments and reaching vaccination sites were generally insignificant, although rural Moldova had appointment scheduling problems. Long waiting times were a significant barrier in urban Serbia. The inability to leave work, school, or childcare duties was a substantial barrier in urban Moldova. There, recommendations from health care providers significantly increased vaccination likelihood and did so even more in rural Moldova. Pleasant interactions with health care providers increased the likelihood of vaccination in urban Serbia and even more so in rural Serbia. Understandable information was a strong motivator in rural Serbia. Descriptive norms, such as knowing most adults were vaccinated, were significant in urban Moldova and urban Serbia. Trust in local health care providers and scientists also emerged as significant motivators, particularly in rural Moldova and rural Serbia.

### 3.2. Design of Behavior Profiles and Strategies

In each country, behavioral profiles were created for the identified priority populations: pregnant women, people with chronic diseases, and health care providers, with the overall goal of increased demand and uptake of COVID-19 vaccination among these priority populations. For pregnant women and people with chronic disease, the priority behaviors were for these populations to receive the full course of COVID-19 vaccination. For health care providers, the priority was to recommend the full course of COVID-19 vaccination to patients (specifically pregnant women and people with chronic disease). In Moldova, Ukrainian refugees were an additional priority population; however, as other aid programs were focused on supporting the refugee population in Moldova, the project prioritized other high-risk populations. Each priority population and behavior are linked with a behavior profile showing the steps, barriers and motivators, key actors, and strategies that form pathways of change. Several strategies emerged from the behavioral profiles including: building institutional capacity for public health information dissemination, fostering partnerships and networks for regional learning exchanges, policies and governance improvements aimed to provide clear guidelines for healthy lifestyles including vaccination, quality improvement initiatives targeting health professional skills, including pre- and in-service training, client service training, and platform use, training sessions for journalists, nongovernmental organizations, nurses, and pharmacists, and organizing community forums and stakeholder reviews to support healthy lifestyle promotion and COVID-19 vaccination. The comprehensive list of recommended strategies is presented in Appendix A.

### 3.3. Program Implementation

#### 3.3.1. Continuing Medical Education (CME) Course

During the implementation period, 107 CME trainings were conducted across the three program countries and included 2403 medical providers (Table 1). In North Macedonia and Serbia, doctors, nurses, and pharmacists were trained; however, in Moldova, only doctors and nurses were included in the trainings. Of the total participants, 1822 responded to the pre-test and 1694 responded to the post-test and CME training evaluation results. Pre-test scores were high, with 88.5% (*n* = 1613) passing by correctly identifying at least six of eight essential pieces of advice about COVID-19 for priority populations. This increased to 99% with the post-test. Based on the post-test and CME evaluation results, 99% (*n* = 1681) of respondents stated that the course improved their knowledge and understanding of healthy lifestyles, immunization, and COVID-19 vaccination. In addition, 97% (*n* = 1646) of participants reported feeling confident or very confident about applying the learning to their work Although pre-test scores were already high, participants found value in the practical and applied aspects of CME training, which they felt improved their understanding and application of the knowledge they had.

In the CME post-evaluation form, participants also reflected on the usefulness of the new teaching techniques. In terms of learning techniques, 32.2% (*n* = 546) preferred case studies; 12.9% (*n* = 219) preferred role plays; and 6.3% (*n* = 106) preferred self-assessments. A majority—78.5% (*n* = 1329)—said a combination of techniques, particularly using case studies and role plays, would most improve their knowledge/understanding of course material.

During qualitative interviews, both CME trainers and participants said that they appreciated the interactive nature of the course, which was different from the passive, lecture-based format typical of most CME courses. For example, one CME trainer said that “The training is very innovative, focusing on interaction rather than classical teaching methods that we are used to. Such an approach motivates you to listen and to participate.”

CME participants similarly noted the value of having time for feedback and discussion during the course because interactions with other participants fostered empathy. For example, in North Macedonia, mixing doctors, nurses, and pharmacists in the training helped build rapport among professionals who typically do not interact. Similarly, participatory activities such as role plays and case studies helped participants consider different perspectives, especially the patient’s. Overall, CME participants said that the participatory training format reaffirmed their knowledge, and the course reinforced and deepened their understanding of the latest evidence about COVID-19 vaccines.

“It was the first time I had the opportunity to participate in role play in this type of course. It is very interesting, you put yourself in the patient’s position and thus you can see your own shortcomings, the skills you lack.”—CME participant, Serbia

Insights from the CME participant interviews and pause-and-reflect sessions with trainers and country facilitators led to adaptations in the CME course content, instructional design, and expansion of the targeted participants. An overview of key adaptations of the CME course is described in Section 3.2.

#### 3.3.2. Collective Engagement Workshops

A total of 267 CE workshops with 3260 chronic disease patients and 842 pregnant women were held across the three countries (Table 2). In qualitative interviews, facilitators noted that discussion was key to building rapport and ensuring that everyone was comfortable talking with each other and asking questions:

“The most useful activities for learning about vaccines against COVID-19 were open discussions, as they gave [participants] the opportunity to express their concerns and receive relevant information.”—CE facilitator, Serbia

“It has a greater effect when we listen to the participants share their experiences than when we tell them stories. The program is good because it is interactive, has games and free discussion.”—CE facilitator, North Macedonia

The role play and puzzle were the most popular workshop activities across all countries because they helped participants discuss COVID-19 concerns and experiences with one another. They also appreciated the discussions, which gave them valuable information and encouraged them to get vaccinated.

“After hearing about the experiences of other participants I am confident that we should be vaccinated against COVID-19.”—CE participant, North Macedonia

“I think that probably discussions after the games were the most helpful. There was a lot of information about the vaccines, and the opportunity to ask if you are not sure or you want to know something more. I also liked the healthy lifestyles we have been talking about, and I find it very useful in my state.”—CE participant, Serbia

“I found group discussions very beneficial, since we could share experiences and discuss vaccination decisions with each other. This created an environment of mutual support and encouragement.”—CE participant, Moldova

Three weeks after each workshop, the project called the 3949 participants who consented to receiving a follow-up call to ask about progress on their immunization commitments. Of the 3494 who answered and agreed to participate in the survey, 38.4% (*n* = 1517) reported that they had set up an appointment with their doctor to re-confirm that they could safely receive the COVID-19 vaccine, and 27.5% (*n* = 960) had sought their partners’ support for vaccination. The project did not observe any meaningful differences in commitment by gender or country.

Roughly 6.75% (*n* = 236) reported receiving a COVID-19 vaccine after the CE workshop. Two hundred of those were from Serbia; 25 from Moldova; and one from North Macedonia. Of these, 48 were pregnant women and 188 were chronic disease patients.

In interviews conducted several weeks after the follow-up calls, the project learned that most participants adhered to their commitments to re-confirm that they could safely receive the vaccine with their doctor or decided to directly visit a vaccination center to receive the vaccine. However, vaccine availability and partner/family support led some to decide not to get vaccinated. This finding led to a key adaptation in the implementation of the CE workshops (described below).

### 3.4. Adaptive Learning during Program Implementation

#### 3.4.1. Overview of Adaptations

In January 2024, based on feedback from CME trainers and CE facilitators from all three countries, the project implemented a series of adaptations to the course and workshop (Table 3 and Table 4, respectively).

#### 3.4.2. Results of the Adaptations

Qualitative interviews with CME trainers and CE facilitators provided insights into how these adaptations are enhancing the practice of recommended behaviors. Specifically, interviews with CME trainers revealed that these adaptations increased participants’ motivation to ask questions and actively participate. They also increased trainers’ ability to establish and maintain the appropriate group dynamic and trust during the course activities. CME participants reported that adaptations increased their use of workshop materials in their work with patients.

CE facilitators said that the provision of tote bags, notepads, and take-home puzzles were an incentive, both because presents evoke positive emotions and notepads are useful for recording workshop information. These materials also reminded participants of the workshops and the commitments they made during them. Inviting participants’ family members and partners to workshops in their own households had multiple benefits. Because people feel safe in their homes, they were more willing to speak openly. It also educated them about the importance of vaccination for family members’/wives’ health, which helped participants fulfill the commitments they made in the workshops. Finally, this approach gave facilitators a better understanding of family dynamics.

## 4. Discussion

During the COVID-19 pandemic, behavioral interventions were utilized to improve demand and uptake of COVID-19 vaccination. A technical brief from the European Centre for Disease Prevention and Control found that behavioral insights strengthened risk communication activities and the direction of messaging around COVID-19 vaccines [22]. Previous research has focused on determinants of COVID-19 vaccination intention, but there is relatively little evidence around interventions targeted at high-risk populations (i.e., pregnant women and older populations living with chronic disease) and how behavioral insights improved their vaccination outcomes [8,23]. This paper is unique in that it outlines the stepwise process taken to design behavior-led strategies focused on both health workers and community members, particularly pregnant women and older populations with chronic disease in the Southeastern European countries of Moldova, North Macedonia, and Serbia. The BI approach helped identify barriers to and motivators of COVID-19 vaccination and develop strategies to increase vaccine uptake. This paper highlights the results from CME courses and CE workshops, including feedback, adaptations, and outcomes.

This project used BI to design and implement a strategy, monitor its activities, and make ongoing adaptations. Behavior change models guide interventions aimed at modifying behaviors. Application and use of BI in this study resulted in the design of an evidence-based, tailored social and behavior change strategy to mitigate barriers to COVID-19 uptake and resulted in increased capacity of health providers to recommend the COVID-19 vaccine and demand for vaccination among priority populations. The use of BI was critical to the success of the intervention, as the stepwise process produced localized and tailored strategies that could be easily implemented in each country.

Vaccine skepticism appears to be more pronounced in Southeastern Europe compared to Western Europe, as evidenced by various studies examining public attitudes towards vaccination, particularly in the context of the COVID-19 pandemic. This skepticism can be attributed to a combination of historical, cultural, and socio-political factors that differ significantly between these regions. Although trust in vaccines varies considerably across Europe, vaccine trust in Northern Europe is at 72%, while it drops to between 32 and 48% in Eastern Europe and is around 59% in Western Europe [24]. Specific determinants of vaccine hesitancy in Eastern European countries are influenced by health and vaccine literacy, which are often lower compared to their Western counterparts [8]. The historical context of distrust in government and health institutions in these regions further exacerbates vaccine skepticism, as individuals may question the motives behind vaccination campaigns [25]. In contrast, Western European countries, despite experiencing their own levels of vaccine hesitancy, generally exhibit higher acceptance rates. For instance, vaccine acceptance rates in Italy and France were reported at 53.7% and 58.9%, respectively, indicating a more favorable attitude towards vaccination compared to the lower rates observed in Southeastern Europe [26]. This difference can be attributed to stronger public health infrastructures and more effective communication strategies regarding vaccine safety and efficacy in Western Europe. Strategies developed using Behavior Integration in our study, which include enhancing public health messaging to build trust, improving health literacy, and engaging the community to advocate for vaccination, show promise in reducing these inequalities. Understanding the variation in vaccine hesitancy across different European regions requires a nuanced approach that considers local contexts and historical backgrounds [9].

Through formative assessment including qualitative and quantitative research, this study identified the main barriers to and motivators of COVID-19 vaccination (Appendix A). This project underlined the importance of health care provider recommendation of the vaccine and the barriers to making such a recommendation, such as lack of time for counseling; fear of harming patients; inability to answer all patient questions, especially when faced with a huge amount of misinformation and disinformation; and general lack of training on communicating about vaccination. In previous studies, communication and vaccine misinformation or disinformation trainings have helped health care providers increase vaccine demand among patients. Health care providers play a critical role in addressing vaccine hesitancy, underlining the importance of identifying hesitant providers, understanding why they are hesitant, and developing tailored strategies to overcome hesitancy so they will build trust with and mitigate patients’ concerns and promote vaccine acceptance [27,28]. In the implementing countries, many medical providers felt ill-equipped to safely refer COVID-19 vaccination to high-risk target populations. Simultaneously, high-risk target populations expressed concern about the safety and efficacy of COVID-19 vaccination because their doctors did not recommend it. This was remedied by the developed behavior change strategy implemented through the CME course and CE workshops.

One challenge the project experienced was growing COVID-19 fatigue among stakeholders who wanted to focus on other public health priorities [29]. Any intervention to improve COVID-19 vaccination demand and uptake would need to highlight both the benefits of vaccination against COVID-19 and those of a healthy lifestyle. The strategy to include COVID-19 in a set of “healthy lifestyle behaviors” through the CME course responded to general health priorities and built basic vaccinology skills and confidence, particularly COVID-19, and taught new techniques and skills in patient care. The facilitation of the CME course through local public health professionals, the Network of Institutes of Public Health (in Serbia), and nongovernmental organizations helped institutionalize the process in communities and beyond, guaranteeing the sustainability of the approach.

The success of community-level programming was significantly attributed to the engagement and link building between community-based organizations and public health institutes. In regions such as Southeastern Europe where outreach from health facilities to communities is less common, community-based organizations were essential because they are trusted. In Serbia, Association Rainbow facilitated CE workshops and conducted follow-up calls with participants. The workshops were a platform for listening to and learning from the communities. The project iterated and adapted the CE workshops to better meet community needs.

Most of our participants in the implementation of the behavior change strategy, including both CME and CE activities, were women. Community-based partners and facilitators noted that recruiting men was a challenge. Our results are therefore skewed to the views and experiences of women in the intervention countries and may not reflect men’s. In general, women participate more frequently in health programs compared to men due to a combination of social, psychological, and structural factors, and even the health care profession attracts predominantly women, especially in Southeast Europe. This may be attributed to women’s generally greater motivation for health-related behavior changes and their willingness to seek support, and social norms and expectations often encourage women to prioritize health and wellness, which can further enhance their participation in health programs [30]. Engaging men in health promotion efforts is crucial, and understanding and addressing the barriers men face in participating in health programs—such as masculine gender roles and social stigma—can help create more inclusive environments [31]. Fostering an environment that encourages collective participation can be beneficial, such as creating supportive networks and communities where both men and women can share experiences and challenges can enhance engagement. In our study, two adaptations were proposed and implemented: workshops for pregnant women including their partners; and workshops for patients with chronic diseases including their family members, carried out in their households in Serbia. The results of these adaptations in our study were very encouraging, with positive reactions from both the CE participants and CE facilitators. Future iterations of the CE workshops could make more effort to recruit and increase men’s engagement and retention into the program.

Health behavior change is a complex process influenced by various factors and stages. This process is evident in COVID-19 vaccination among hesitant populations, where individuals may progress through different stages of behavior change before accepting vaccination. Observing stepwise progression among community participants toward full vaccination, the project aimed to understand motivators and barriers that CE workshop participants experienced. The interviews conducted several weeks after the final follow-up calls revealed that most participants moved successfully along this stepwise pathway and followed through on commitments made during the CE workshops. However, some participants continued to experience obstacles to vaccination (e.g., vaccines were unavailable, partners or family members were not supportive) that hindered their ability to achieve the main behavior of getting fully vaccinated against COVID-19.

There are several obvious limitations to this study, mainly originating from the non-randomized and non-controlled characteristics of its design. The formative assessment from which this paper draws had varied timelines and was influenced by external variables (political situation, vaccine availability, changing COVID-19 landscape), and its varied methodologies were designed to answer specific programmatic questions rather than test a common hypothesis, limiting the ability to conduct a rigorous comparative analysis across all parameters. The effectiveness of the behavior change strategy could not be compared to populations that were not exposed to the project’s social and behavior change intervention. The outcomes of the intervention (commitment to make an appointment with the doctor, discuss COVID-19 vaccination, and get vaccinated) are self-reported but indicate that steps toward vaccination occurred. Finally, CME training and CE workshop feedback demonstrated rebuilt trust and a positive experience for all involved. Future research should focus on three key areas of improvement: (a) controlled comparative studies, such as randomized controlled trials, to rigorously evaluate the effectiveness of behavior change strategies; (b) longitudinal studies to assess the long-term impact of interventions, utilizing objective measures such as actual vaccination records; and (c) integration of adaptive learning in implementation, focusing on how iterative feedback and real-time adjustments to interventions can enhance the effectiveness of behavior change strategies.

The health care systems of Moldova, Serbia, and North Macedonia share several key similarities, particularly in their structure and challenges. All three countries have predominantly public health care systems, funded largely through national health insurance schemes that aim to provide universal health coverage. Primary health care serves as the foundation of their health systems, with a focus on providing access to essential services such as immunization and maternal care. However, these systems face common challenges, including limited financial resources and regional disparities in health care access and availability of health care professionals [32,33,34]. Furthermore, all three countries have been significantly impacted by the COVID-19 pandemic, which has exacerbated existing challenges related to vaccine distribution, health infrastructure, trust, and public health communication. The approach proposed in this project could be adapted and applied to countries with similar health care systems’ organization and challenges, especially in Eastern and Southeastern Europe.

## 5. Conclusions

The use of BI as the behavior change framework can lead to the development of evidence-based, tailored strategies that significantly boosted vaccine demand and confidence. This approach, combined with adaptive learning processes, enables projects to refine program implementation to meet changing needs. Stakeholders’ continued involvement is crucial for developing relevant materials and activities; therefore, collaborating with local implementing organizations and community-based groups ensures a project’s sustainability and responsiveness to local contexts.

Promoting acceptance by framing COVID-19 vaccination as part of a healthy lifestyle (integrating with other priorities), using existing health care providers as trainers, and the adaptability of the developed CME and CE materials to other diseases and vaccines enhances a program’s utility and impact, ensuring health care workers are well equipped to recommend vaccinations across the life course. Strengthening collaboration between community-based organizations and health facilities and providing all developed materials to public health professionals, doctors, and community-based organization staff ensures that these resources will support future immunization campaigns and public health initiatives.

Finally, to enhance emergency preparedness, it is critical to build health care providers’ skills in vaccinology and communication well before public health crises arise. This should involve integrating comprehensive vaccine education at all levels of medical training—undergraduate, postgraduate, and continuing medical education—to ensure that health care professionals are equipped with the necessary knowledge to address vaccine-related issues during routine care and emergencies alike. Additionally, public education on vaccines must be strengthened outside of emergency contexts to foster trust and understanding. Closing these knowledge gaps proactively, rather than relying on reactive measures during crises, will improve routine vaccination uptake and ensure a more resilient health care system during future emergencies.

## Figures and Tables

**Figure 1 vaccines-12-01131-f001:**
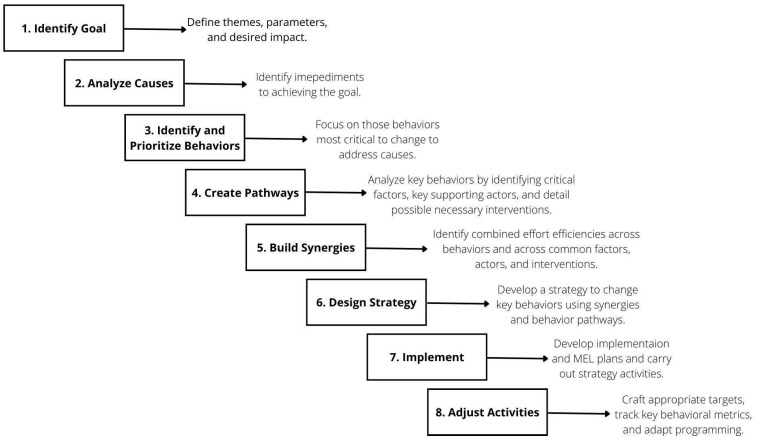
The Behavior Integration Guidance Process Schema.

**Figure 2 vaccines-12-01131-f002:**
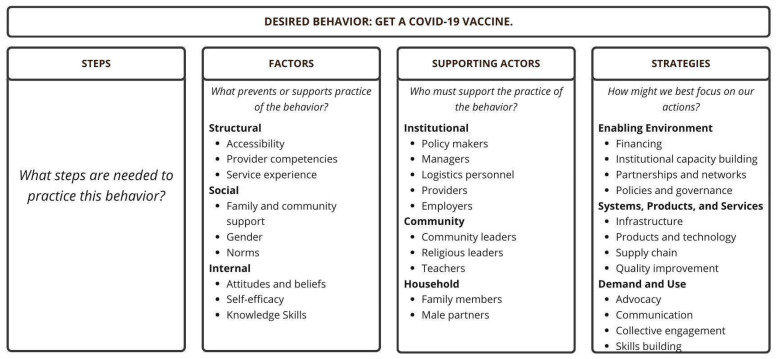
Behavior Integration Guidance: Mapping Pathways to Change—The Behavior Profile.

**Table 1 vaccines-12-01131-t001:** CME Trainings and Participants.

Country	Trainings Completed	Total Participants by gender
North Macedonia	32	549 (162 M/387 F)
Moldova	10	192 (34 M/158 F)
Serbia	65	1662 (288 M/1374 F)
Total	107	2403 (871 M/1919 F)

**Table 2 vaccines-12-01131-t002:** CE Workshop Types and Participants, by Country.

	Serbia	North Macedonia	Moldova	Total
Priority Group	Number of Workshops			
Pregnant women	34	0	21	55
Chronic disease patients	120	71	21	212
Total	154	71	42	267
Priority Group	Number of Participants			
Pregnant women	475	N/A	367	842
Chronic disease patients	1836 (478 M/1358 F)	1050 (307 M/743 F)	374 (65 M/309 F)	3260 (850 M/2410 F)
Total	2311	1050	741	4102

**Table 3 vaccines-12-01131-t003:** CME Course Feedback and Adaptations.

Feedback	Adaptation
Inability of medical professionals to participate in a day-long CME course.	Created a modular version of the curriculum with optional activities.
Ministry of Health interest in vaccines beyond COVID-19.	Added information on the flu vaccine and measles vaccination, due to outbreaks in Moldova.
CME trainers and CE workshop facilitators wanted more information about facilitation techniques.	Developed an additional frequently asked questions document with more information on facilitation techniques.

**Table 4 vaccines-12-01131-t004:** CE Workshop Feedback and Adaptations.

Feedback	Adaptation
The puzzle activity was an effective and diverse discussion starter. It elicited “strong emotions”, “surprise”, and “nostalgic memories” among participants.	Tote bags with puzzles and notepads given to participants in Serbia.
Decision making often involves people beyond the individual to be vaccinated.	Conducted five pilot workshops with pregnant women and their partners and five workshops with chronic disease patients and their family members in their households in Serbia.
SMS reminders were too frequent in North Macedonia.	Decreased frequency of SMS reminders from seven times to once a week and modified message content.
Facilitators wanted more information about facilitation techniques.	Developed an additional frequently asked questions document with more information on facilitation techniques.

## Data Availability

Data are available upon reasonable request to the corresponding author or the JSI, Arlington, VA, USA.

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
