# Peer review of "Using Behavior Integration to Identify Barriers and Motivators for COVID-19 Vaccination and Build a Vaccine Demand and Confidence Strategy in Southeastern Europe"

_vaccines, 2024, doi:10.3390/vaccines12101131_

Round 1

Reviewer 1 Report

Comments and Suggestions for Authors

Dear Authors,

I thank you for your research efforts. The article is both engaging and well-composed. However, I would like to highlight a concern regarding the sample of participants: it is not stratified by age, and it lacks gender balance, with unequal representation of men and women. This limitation should be explicitly acknowledged in the relevant section of the paper.

Additionally, I recommend enhancing the introduction by addressing the broader challenges posed by the pandemic, which extend beyond economic issues to include ethical considerations. To this end, you might find it beneficial to read some articles, for example "Ethical Criteria for the Admission and Management of Patients in the ICU Under Conditions of Limited Medical Resources: A Shared International Proposal in View of the COVID-19 Pandemic" (PMID 32612972).

Best regards

Author Response

Dear Colleague,

We are grateful for your attention to our Manuscript and the constructive comments you have provided us to improve it. We have answered each of your individual comments (your comments are in BOLD), and provided you with the change we propose in the text of the Manuscript. We remain open to any additional suggestions for improvements.

I thank you for your research efforts. The article is both engaging and well-composed.

However, I would like to highlight a concern regarding the sample of participants: it is not stratified by age, and it lacks gender balance, with unequal representation of men and women. This limitation should be explicitly acknowledged in the relevant section of the paper.

### Thank you for underlining this limitation of our research, and luckily we have been discussing this issue during our project’s implementation. Women, in general, are more focused on health and are more likely to seek support and utilize healthcare services more. In most areas of the world, South East Europe and countries included in our research being a great example, women are those who take care of the health of the family, and are even “expected” to participate in health programs. We have addressed this in the discussion/limitations of our paper now explicitly, and we provide some recommendations on how to overcome this gap.

— Added in Discussion: Most of our participants in the implementation of the behavior change strategy, including both CME and CE activities, were women. In general, women participate more frequently in health programs compared to men due to a combination of social, psychological, and structural factors, and even the healthcare profession attracts predominantly women, especially in South East Europe. This may be attributed to women's generally greater motivation for health-related behavior changes and their willingness to seek support, and social norms and expectations often encourage women to prioritize health and wellness, which can further enhance their participation in health programs (Kothari et al., 2014). Engaging men in health promotion efforts is crucial, and understanding and addressing the barriers men face in participating in health programs—such as masculine gender roles and social stigma—can help create more inclusive environments (Howell et al., 2022). Fostering an environment that encourages collective participation can be beneficial, such as creating supportive networks and communities where both men and women can share experiences and challenges can enhance engagement. In our study, two adaptations were proposed and implemented: workshops for pregnant women including their partners; and workshops for patients with chronic diseases including their family members, and done in their household in Serbia. The results of these adaptations in our study were very encouraging, with positive reactions from both the CE participants and CE facilitators.

Howell, B., Peterson, J., & Corbett, S. (2022). Where are all the men? A qualitative review of the barriers, facilitators, and recommendations to older male participation in health promotion interventions. American Journal of Health Promotion, 37(3), 386-400. https://doi.org/10.1177/08901171221123053

Kothari, C., Butkiewicz, R., Williams, E., Jacobson, C., Morse, D., & Cerulli, C. (2014). Does gender matter? exploring mental health recovery court legal and health outcomes. Health & Justice, 2(1). https://doi.org/10.1186/s40352-014-0012-0

Additionally, I recommend enhancing the introduction by addressing the broader challenges posed by the pandemic, which extend beyond economic issues to include ethical considerations. To this end, you might find it beneficial to read some articles, for example "Ethical Criteria for the Admission and Management of Patients in the ICU Under Conditions of Limited Medical Resources: A Shared International Proposal in View of the COVID-19 Pandemic" (PMID 32612972).

### Thank you for this comment and suggestion. We have evaluated the proposed paper and included the importance of thinking about the common good, which reflects well on the behavior change approach which also focuses on the community, the individual, and the relationship between them.

— In the Introduction: Along with economic issues the pandemic has also posed ethical challenges, where one of the key questions is how to achieve the common good. The principle of the common good implies that what is good for society is also good for the individual. Furthermore, from the Global Health perspective that means that patients cannot be perceived as isolated human beings but persons strongly bound with their communities, suggesting that both patients and the community need to be taken in consideration

Reviewer 2 Report

Comments and Suggestions for Authors

Dear Authors,

Your article is very interesting.

The article is written in a scientific way. Its results are important for the stakeholders in Eastern Europe. It adds knowledge on how to increase vaccine coverage in the region. Eastern Europe is different from West and Central Europe so such research is necessary. The knowledge gap could be filled in by qualitative research like this.  

However I have several comments:

I would suggest removing the hyperlinks within the text. If necessary, you can add a reference or footnote.

One more suggestion: to use the symbol % instead of “percent”.

Author Response

We thank the Reviewer for their kind comments and suggestions for the improvement of our Manuscript. We have addressed each comment (your comments are in BOLD) with adequate changes in the text, and we remain open to any further suggestions for improvement.

Your article is very interesting.

The article is written in a scientific way. Its results are important for the stakeholders in Eastern Europe. It adds knowledge on how to increase vaccine coverage in the region. Eastern Europe is different from West and Central Europe so such research is necessary. The knowledge gap could be filled in by qualitative research like this.

However I have several comments:

I would suggest removing the hyperlinks within the text. If necessary, you can add a reference or footnote.

### We have removed hyperlinks from the text and put them in footnotes. We will modify further in case the technical process of the Journal requires it.

One more suggestion: to use the symbol % instead of “percent”.

### We have replaced the word “percent” with the symbol “%” throughout the text.

Reviewer 3 Report

Comments and Suggestions for Authors

Not very clear why these three countries were chosen for the study? the health systems in Serbia and North Macedonia are much more close to each other due to historical background and geography, Moldova is certainly somewhat different (except GDP being close to NM). 

The use of term "Eastern Europe" is definitively not a good ideas as it is largely ambiguous term, it has a wide range of various denominator depending connotations. I would recommend that you define what you mean under "Eastern Europe" in terms countries involved. I would suggest  using Southeast Europe/Southeastern Europe (although not ideal as well) be even a better choice bringing the focus more where it actually is. Or be even more precise - just mentioning the 3 countries. 

In the introduction you give the impression that "Eastern European" (EE) countries have higher prevalence of vaccine skepticism. In spite of not knowing the countries you involve in the EE I would dare to say that France from the EU has likely more vaccine skepticism than many EE countries. I recall some studies have called it the most vaccines skeptic country in the Europe. And this already before COVID-19. I do not think it magically improved with COVID-19.   

In the discussion of results you indirectly point out (without saying it) that knowledge caps in health care providers education did not enable them to make solid credible recommendations for vaccination(s). 

Should one of the recommendation be suggesting closing these knowledge caps about vaccines in more general terms rather than just citing project created tools? It would be good to learn lessons from COVID-19 and improve the education regarding vaccines of health care providers on all levels - undergraduate, postgraduate and CME? It is also clear that better education of the public is needed. All this should NOT be done as campaign when we have the emergency. Your project was certainly great success but the best would be that we do not NEED this type of projects. We need to close the knowledge caps BEFORE emergencies and for the success of routine vaccinations needed.  

 The whole article,  especially some parts of it (results), is a bit lengthy. Obviously it is a very good project report, but not necessarily the ideal succinct scientific paper. Also the recommendations are very much keeping close link to the project. Thus, they are not easily generalizable to other countries beyond the 3 project countries.  

Author Response

Dear Reviewer,

We would like to thank you for dedicating your time to a detailed review of our Manuscript. We have gladly accepted all your suggestions for improvement and have incorporated them in our new version of the Manuscript. We remain open to any further suggestions and thank you again for helping us improve our Manuscript.

Below, you see your comments (in BOLD) with answers from our research team and various changes we have incorporated into our Manuscript.

Not very clear why these three countries were chosen for the study? the health systems in Serbia and North Macedonia are much more close to each other due to historical background and geography, Moldova is certainly somewhat different (except GDP being close to NM).

### Thank you for underlining this potential issue. You are right that Serbia and North Macedonia share more similarities regarding the language, history, and of course a (previously united) healthcare system. Nevertheless, the healthcare systems of Moldova, Serbia, and North Macedonia share several key similarities, particularly in their structure and challenges. All three countries have predominantly public healthcare systems, funded largely through national health insurance schemes that aim to provide universal health coverage. Primary healthcare serves as the foundation of their health systems, with a focus on providing access to essential services such as immunization and maternal care. However, these systems face common challenges, including limited financial resources, regional disparities in healthcare access, and a shortage of healthcare professionals. Furthermore, all three countries have been significantly impacted by the COVID-19 pandemic, which has exacerbated existing challenges related to vaccine distribution, health infrastructure, and public health communication. We have added this information to the Discussion, which has also helped us improve the recommendations beyond just the three countries. On the other hand, the three countries were chosen due to a common project, which we have tried to state more clearly in the Introduction and Methods sections of the Manuscript.

— In Introduction: USAID-funded MOMENTUM Routine Immunization Transformation and Equity (the Project) began work in South Eastern European countries of Moldova, North Macedonia and Serbia in May 2022.

— In Methods: USAID’s Europe and Eurasia Regional Bureau asked the Project to accelerate widespread and equitable access to and delivery of safe and effective COVID-19 vaccinations by increasing demand for and uptake of COVID-19 vaccination among priority populations in Moldova, North Macedonia, and Serbia. The project’s scope was to: 1) increase demand for COVID-19 vaccination; 2) dispel myths and mis/disinformation; 3) implement behavior change interventions; and 4) share lessons and promising practices within and between Moldova, North Macedonia, and Serbia, and across the region by using BI to develop tailored health professional capacity building and collective engagement (CE) programming in each country.

— In Discussion: The healthcare systems of Moldova, Serbia, and North Macedonia share several key similarities, particularly in their structure and challenges. All three countries have predominantly public healthcare systems, funded largely through national health insurance schemes that aim to provide universal health coverage. Primary healthcare serves as the foundation of their health systems, with a focus on providing access to essential services such as immunization and maternal care. However, these systems face common challenges, including limited financial resources, and regional disparities in healthcare access and availability of healthcare professionals (Bjegovic-Mikanovic et al., 2019; Milevska Kostova et al., 2017; Turcanu et al., 2012). Furthermore, all three countries have been significantly impacted by the COVID-19 pandemic, which has exacerbated existing challenges related to vaccine distribution, health infrastructure, trust, and public health communication. The approach proposed in this project could be adapted and applied to countries with similar healthcare systems’ organization and challenges, especially in Eastern and South Eastern Europe.

Turcanu, Ghenadie, Silviu Domente, Mircea Buga, Erica Richardson, and World Health Organization. "Republic of Moldova: Health system review." (2012).

Bjegovic-Mikanovic, Vesna, Milena Vasic, Dejana Vukovic, Janko Jankovic, Aleksandra Jovic-Vranes, Milena Santric-Milicevic, Zorica Terzic-Supic, Cristina Hernández-Quevedo, and World Health Organization. "Serbia: Health system review." (2019).

Milevska Kostova, Neda, Snezhana Chichevalieva, Ninez A. Ponce, Ewout van Ginneken, Juliane Winkelmann, and World Health Organization. "The former Yugoslav Republic of Macedonia: health system review." (2017).

The use of term "Eastern Europe" is definitively not a good ideas as it is largely ambiguous term, it has a wide range of various denominator depending connotations. I would recommend that you define what you mean under "Eastern Europe" in terms countries involved. I would suggest using Southeast Europe/Southeastern Europe (although not ideal as well) be even a better choice bringing the focus more where it actually is. Or be even more precise - just mentioning the 3 countries.

### Thank you for pointing this out, and we agree, and have replaced Eastern Europe with Southeastern Europe in the title and throughout the text. We agree that this is better, although not ideal, as some countries are placed in other geographical categories, depending on the context.

— Changed throughout the text where the term is related to the countries participating in the study.

In the introduction you give the impression that "Eastern European" (EE) countries have higher prevalence of vaccine skepticism. In spite of not knowing the countries you involve in the EE I would dare to say that France from the EU has likely more vaccine skepticism than many EE countries. I recall some studies have called it the most vaccines skeptic country in the Europe. And this already before COVID-19. I do not think it magically improved with COVID-19.

### Thank you for this comment, and we find it very unfortunate that in this age we are forced to compare our vaccine skepticism between countries, instead of taking positive examples from each country and applying them to improve populations’ health. We have looked for relevant references to help us improve and better discuss this apparent inequality, and we provide it in the Discussion section now.

— In Discussion: Vaccine skepticism appears to be more pronounced in Southeastern Europe compared to Western Europe, as evidenced by various studies examining public attitudes towards vaccination, particularly in the context of the COVID-19 pandemic. This skepticism can be attributed to a combination of historical, cultural, and socio-political factors that differ significantly between these regions. Although trust in vaccines varies considerably across Europe, vaccine trust in Northern Europe was at 72%, while it drops to between 32-48% in Eastern Europe and is around 59% in Western Europe (Tetik et al. 2022). Specific determinants of vaccine hesitancy in Eastern European countries are influenced by health and vaccine literacy, which are often lower compared to their Western counterparts (Popa et al., 2022). The historical context of distrust in government and health institutions in these regions further exacerbates vaccine skepticism, as individuals may question the motives behind vaccination campaigns (Cascini et al., 2021). In contrast, Western European countries, despite experiencing their own levels of vaccine hesitancy, generally exhibit higher acceptance rates. For instance, vaccine acceptance rates in Italy and France were reported at 53.7% and 58.9%, respectively, indicating a more favorable attitude towards vaccination compared to the lower rates observed in Southeastern Europe (Sallam, 2021). This difference can be attributed to stronger public health infrastructures and more effective communication strategies regarding vaccine safety and efficacy in Western Europe. Strategies developed using Behavior Integration in our study, which include enhancing public health messaging to build trust, improving health literacy, and engaging the community to advocate for vaccination, show promise in reducing these inequalities. Understanding the variation in vaccine hesitancy across different European regions requires a nuanced approach that considers local contexts and historical backgrounds (Toshkov, 2023).

Cascini, F., Pantović, A., Al-Ajlouni, Y., Failla, G., & Ricciardi, W. (2021). Attitudes, acceptance and hesitancy among the general population worldwide to receive the covid-19 vaccines and their contributing factors: a systematic review. Eclinicalmedicine, 40, 101113. https://doi.org/10.1016/j.eclinm.2021.101113

Popa, A., Enache, A., Popa, I., Antoniu, S., Dragomir, R., & Burlacu, A. (2022). Determinants of the hesitancy toward covid-19 vaccination in eastern european countries and the relationship with health and vaccine literacy: a literature review. Vaccines, 10(5), 672. https://doi.org/10.3390/vaccines10050672

Sallam, M. (2021). Covid-19 vaccine hesitancy worldwide: a concise systematic review of vaccine acceptance rates. Vaccines, 9(2), 160. https://doi.org/10.3390/vaccines9020160

Tetik, B., Tekin, Ç., Delen, L., Tekinemre, I., & Bayindir, Y. (2022). Experience and opinions of healthcare professionals on covid-19 and inactive covid-19 vaccine (coronavac, developed by sinovac of china). Medicine Science | International Medical Journal, 11(3), 1494. https://doi.org/10.5455/medscience.2022.05.121

Toshkov, D. (2023). What accounts for the variation in covid-19 vaccine hesitancy in eastern, southern and western europe?. Vaccine, 41(20), 3178-3188. https://doi.org/10.1016/j.vaccine.2023.03.030

In the discussion of results you indirectly point out (without saying it) that knowledge caps in health care providers education did not enable them to make solid credible recommendations for vaccination(s).

Should one of the recommendation be suggesting closing these knowledge caps about vaccines in more general terms rather than just citing project created tools? It would be good to learn lessons from COVID-19 and improve the education regarding vaccines of health care providers on all levels - undergraduate, postgraduate and CME? It is also clear that better education of the public is needed. All this should NOT be done as campaign when we have the emergency. Your project was certainly great success but the best would be that we do not NEED this type of projects. We need to close the knowledge caps BEFORE emergencies and for the success of routine vaccinations needed.

### Thank you for this comment, and we couldn’t agree more. We have prepared an additional paragraph for the Conclusion of the paper underlining exactly that.

— In Conclusions, last paragraph: Finally, to enhance emergency preparedness, it is critical to build healthcare providers' skills in vaccinology and communication well before public health crises arise. This should involve integrating comprehensive vaccine education at all levels of medical training—undergraduate, postgraduate, and continuing medical education (CME)—to ensure that healthcare professionals are equipped with the necessary knowledge to address vaccine-related issues during routine care and emergencies alike. Additionally, public education on vaccines must be strengthened outside of emergency contexts to foster trust and understanding. Closing these knowledge gaps proactively, rather than relying on reactive measures during crises, will improve routine vaccination uptake and ensure a more resilient healthcare system during future emergencies.

The whole article,  especially some parts of it (results), is a bit lengthy. Obviously it is a very good project report, but not necessarily the ideal succinct scientific paper. Also the recommendations are very much keeping close link to the project. Thus, they are not easily generalizable to other countries beyond the 3 project countries.

### We have tried to respond to this request, although it is not an easy task. We have had many discussions among out team regarding the content to include in this Manuscript. It was our goal to provide a full overview of all phases of our work and their outcomes, but to also include the information about all the tools/processes we used and documented, to allow a future research team to apply the same process with all the necessary information in one place. This resulted in a lengthy Manuscript. On the other hand, with the help of comments you and other reviewers provided, we believe to have made our results more relevant and generalizable to other countries finding themselves in a similar context, as well as developed recommendations which could improve emergency preparedness in any country. We remain open to any additional comments or suggestions which could improve our Manuscript.

Reviewer 4 Report

Comments and Suggestions for Authors

Thank you for inviting me to review this manuscript. I appreciate that the authors did a solid work collecting responses from different countries. The findings of such work would be important in future pandemics when vaccination is necessary. However, I'm not sure whether this is a scientific manuscript for journal publication or a master's thesis/PhD dissertation with so many supplementary materials reaching 11 documents, an extremely long methods section that even includes some results, and tables that include nothing about the results of the quantitative data collection (no demographics of the participants and descriptive statistics of their responses to key questions). Also, all the tables are only focused on the CME part of the project that is mentioned late in the methods section. The figures are okay and I think they can replace some of the supplementary materials as the authors see fit. The authors need to fully revise their work to make it more concise, readable, reproducible, and applicable to practice.

Some specific issues are outlined below:
1. Abstract: Change "percent" to "%"
2. I suggest adding "Eastern Europe" to the list of keywords.
3. Line 65: Spell out "USAID"
4. Lines 99-103: Since it has been a long while since the COVID-19 pandemic, I suggest that the authors add to their objective the implications of the findings of this study on future pandemics when vaccination is necessary.
5. Under "Qualitative data collection", how were the participants recruited? What recruitment method did you employ and where?
6. Under "Quantitative data collection", was the online survey validated? Was it tested for test retest reliability? Did you have a pilot phase to test it before distributing it?
7. Line 191: Remove "and"
8. The entire methods section is extremely long. It even includes some results! (see line 230 and forward). Please summarize your methods and remove any results from it.
9. The discussion section mainly focused on re-summarizing the findings of the study. It lacked comparing the authors' results with results from previous similar studies. There is a plenty of studies in the literature on COVID-19 vaccine uptake (including facilitators and barriers) in Eastern Europe. I suggest the authors run a literature search and cite relevant studies accordingly in line with their findings.
10. The conclusion is good. It's concise, clear, and includes future recommendations.

Author Response

Dear Reviewer,

Thank you for taking the time and providing us with detailed comments and suggestions on how to improve our Manuscript. Based on your comments and suggestions, together with some comments of other Reviewers which we addressed, we have done our best to improve the Manuscript and make it more readable and relevant for the audience.

We have had many discussions among out team regarding the content to include in this Manuscript. It was our goal to provide a full overview of all phases of our work and their outcomes, but to also include information about all the tools/processes we used and documented, to allow a future research team to apply the same process with all the necessary information in one place. This resulted in a lengthy Manuscript. On the other hand, with the help of comments you and other reviewers provided, we believe to have made our results more relevant and generalizable to other countries finding themselves in a similar context, as well as developing recommendations that could improve emergency preparedness in any country.

Below, we answer your comments one by one (your comments are in BOLD), and provide the proposed change and location in the text where the change can be found. We remain open to any additional comments or suggestions that could improve our Manuscript.

Thank you for inviting me to review this manuscript. I appreciate that the authors did a solid work collecting responses from different countries. The findings of such work would be important in future pandemics when vaccination is necessary. However, I'm not sure whether this is a scientific manuscript for journal publication or a master's thesis/PhD dissertation with so many supplementary materials reaching 11 documents, an extremely long methods section that even includes some results, and tables that include nothing about the results of the quantitative data collection (no demographics of the participants and descriptive statistics of their responses to key questions). Also, all the tables are only focused on the CME part of the project that is mentioned late in the methods section. The figures are okay and I think they can replace some of the supplementary materials as the authors see fit. The authors need to fully revise their work to make it more concise, readable, reproducible, and applicable to practice.

### Thank you for your valuable feedback. We agree that some of the results could be better placed in the Results section, and we have made the necessary adjustments to ensure clarity and moved everything that could be considered a “result” of our work to the Results section. However, we would like to clarify that the primary focus of this paper is the process and methodology we employed to identify barriers and motivators for vaccination, the approach to creating an SBC strategy (using BI), implementing the strategy and evaluating it, rather than providing a comprehensive analysis of the quantitative data itself. Our goal is to demonstrate the approach of using both primary and secondary data collection to inform behavior profiles, with quantitative data serving to support and validate the process. We hope to have a chance to publish a more detailed analysis of the quantitative findings in a separate manuscript at some point. We believe this approach allows us to focus on the critical steps involved in behavioral integration, while ensuring that more in-depth statistical analysis receives appropriate attention in future publications. Thank you again for your insightful comments, which have helped to improve the structure and clarity of our paper. We have, however, prepared additional descriptive information regarding the population included in the quantitative data collection, and we provide this information in the revised version of the Manuscript.

— In Results (3.1. Formative assessment results - Quantitative data): In Moldova, the study population consisted of 1,412 participants, with a slight majority being female (54.0%) compared to males (43.1%). The age distribution revealed that nearly half (47.2%) were between 18-25 years, while 23.5% were aged 26-35 years. Most participants resided in city centers or metropolitan areas (46.0%), with a significant portion from rural areas (34.6%). The education levels were diverse, with 39.4% holding secondary education and 30.0% possessing a university or college degree. Employment status indicated a large proportion of students (37.0%), followed by full-time employed individuals (24.6%). The predominant ethnic group was Moldovan (73.8%), with smaller representations of Russian (12.3%), Romani (4.0%), and Ukrainian (5.0%) minorities.

In Serbia, there were 1,112 participants, with females comprising the majority (60.2%) and males representing 38.8%. The largest age group was also 18-25 years (33.5%), followed by participants aged 26-35 years (26.0%). Nearly half (45.7%) resided in city centers or metropolitan areas, while 37.5% were from suburban or peri-urban areas. The majority had completed secondary education (61.8%), with 23.6% holding university or college degrees. Employment data showed 47.6% were full-time employed, and 22.1% were students. The vast majority of participants identified as Serbian (93.2%), with smaller ethnic groups such as Romani, Russian, and Bulgarian collectively making up less than 5%.

Some specific issues are outlined below:
1. Abstract: Change "percent" to "%"

### Changed throughout the Manuscript.

  1. I suggest adding "Eastern Europe" to the list of keywords.

### Based on another reviewer’s suggestion, we have changed the title to read “Southeastern Europe” to better reflect the geographical position of countries involved. We have added “Southeastern Europe” to the list of keywords.

  1. Line 65: Spell out "USAID"

### Spelled out on the first mention.

  1. Lines 99-103: Since it has been a long while since the COVID-19 pandemic, I suggest that the authors add to their objective the implications of the findings of this study on future pandemics when vaccination is necessary. ?

### We have focused our discussion more on the general vaccine skepticism in the region, through and outside of COVID-19, and on the need of having a strategy to improve healthcare providers’ knowledge of vaccinology and communication skills as regular activities. Our Discussion and Conclusions now reflect these changes.

— In Conclusions: Finally, to enhance emergency preparedness, it is critical to build healthcare providers' skills in vaccinology and communication well before public health crises arise. This should involve integrating comprehensive vaccine education at all levels of medical training—undergraduate, postgraduate, and continuing medical education (CME)—to ensure that healthcare professionals are equipped with the necessary knowledge to address vaccine-related issues during routine care and emergencies alike. Additionally, public education on vaccines must be strengthened outside of emergency contexts to foster trust and understanding. Closing these knowledge gaps proactively, rather than relying on reactive measures during crises, will improve routine vaccination uptake and ensure a more resilient healthcare system during future emergencies.

  1. Under "Qualitative data collection", how were the participants recruited? What recruitment method did you employ and where?

### Thank you for this question. Purposeful sampling was applied, and stakeholders were selected based on their professional experience and particular knowledge which allowed a deeper insight in all the aspects relevant to the study.

— This is now reflected in the Methods section.

  1. Comment: Under "Quantitative data collection", was the online survey validated? Was it tested for test retest reliability? Did you have a pilot phase to test it before distributing it?

### The online survey was developed using the validated Behavioral and Social Drivers of Vaccination toolkit developed by the WHO with questions reviewed and agreed upon by experts in the field. The online survey was piloted in both Serbia and Moldova with 100 participants in each country. In the pilot, we targeted unvaccinated individuals (50 per country) and health workers (25 per country). The team updated the survey based on the responses received during the pilot phase. The survey itself was not tested for test/retest reliability.

— In Methods (Quantitative data collection): In Moldova and Serbia, the project adapted questions from the validated Behavioral and Social Drivers for Vaccination toolkit to develop an online survey tool administered through the crowdsourcing platform PREMISE. The tool was piloted in both countries with 100 participants in each country. In the pilot, unvaccinated individuals (50) and health workers (25) were the main targets of the data collection. After revising the tool based on feedback from the pilot, the project collected data from 2,742 participants living in Serbia and Moldova

  1. Line 191: Remove "and"

### Removed and rephrased the sentence.

  1. The entire methods section is extremely long. It even includes some results! (see line 230 and forward). Please summarize your methods and remove any results from it.

### We have reduced the Method section and moved the results which were presented into the Results section of the paper. We still believe that the Supplementary Materials allow for the reproducibility of our work and would like to share all the additional documents with the Manuscript to allow for wide access.

  1. The discussion section mainly focused on re-summarizing the findings of the study. It lacked comparing the authors' results with results from previous similar studies. There is a plenty of studies in the literature on COVID-19 vaccine uptake (including facilitators and barriers) in Eastern Europe. I suggest the authors run a literature search and cite relevant studies accordingly in line with their findings.

### Thank you for this comment. Based on your comment and that of other reviewers we have completely reframed the Discussion of our paper to focus on the novelty of our approach and to cover the comparison with previous similar studies. We also identify some gaps in the literature and future directions.

— In Discussion: During the COVID-19 pandemic, behavioral interventions were utilized to improve demand and uptake of COVID-19 vaccination. A technical brief from the European Centre for Disease Prevention and Control found that behavioral insights strengthened risk communication activities and the direction of messaging around COVID-19 vaccines. Several publications have focused on determinants of COVID-19 vaccination intention, but there is relatively little evidence around interventions targeted at high-risk populations (i.e. pregnant women and older populations living with chronic disease) and how behavioral insights improved their vaccination outcomes. This paper is unique in that it outlines the stepwise process taken to design behavior-led strategies focused on both health workers and community members, particularly pregnant women and older populations with chronic disease.

  1. The conclusion is good. It's concise, clear, and includes future recommendations.

### Thank you! We have added another paragraph to the conclusions regarding the importance of working with healthcare providers continuously and at all levels of education, as part of emergency preparedness. This was suggested by one of the other reviewers and fits well with the importance of health providers’ recommendations for vaccine uptake.

— In Conclusions: Finally, to enhance emergency preparedness, it is critical to build healthcare providers' skills in vaccinology and communication well before public health crises arise. This should involve integrating comprehensive vaccine education at all levels of medical training—undergraduate, postgraduate, and continuing medical education (CME)—to ensure that healthcare professionals are equipped with the necessary knowledge to address vaccine-related issues during routine care and emergencies alike. Additionally, public education on vaccines must be strengthened outside of emergency contexts to foster trust and understanding. Closing these knowledge gaps proactively, rather than relying on reactive measures during crises, will improve routine vaccination uptake and ensure a more resilient healthcare system during future emergencies.

Round 2

Reviewer 4 Report

Comments and Suggestions for Authors

Many thanks to the authors for addressing the comments and improving their manuscript. I have no further comments.